# IL-31RA and TRPV1 Expression in Atopic Dermatitis Induced with Trinitrochlorobenzene in Nc/Nga Mice

**DOI:** 10.3390/ijms241713521

**Published:** 2023-08-31

**Authors:** Seokwoo Lee, Na Yeon Lim, Min Soo Kang, Yunho Jeong, Jin-Ok Ahn, Jung Hoon Choi, Jin-Young Chung

**Affiliations:** 1Department of Veterinary Internal Medicine and Institute of Veterinary Science, College of Veterinary Medicine, Kangwon National University, 1 Kangwondaehak-gil, Chuncheon-si 24341, Gangwon-do, Republic of Korea; thou12@nate.com (S.L.); mignon0729@gmail.com (N.Y.L.); jsteve35@outlook.com (Y.J.); joahn@kangwon.ac.kr (J.-O.A.); 2Department of Veterinary Anatomy and Institute of Veterinary Science, College of Veterinary Medicine, Kangwon National University, 1 Kangwondaehak-gil, Chuncheon-si 24341, Gangwon-do, Republic of Korea; imkangms93@kbsi.re.kr (M.S.K.); jhchoi@kangwon.ac.kr (J.H.C.)

**Keywords:** atopic dermatitis, IL-31, Nc/Nga mouse, sensory nerve fiber, trinitrochlorobenzene, TRPV1

## Abstract

Atopic dermatitis (AD) is a common chronic inflammatory skin disease. Interleukin 31 (IL-31), a novel cytokine in AD, causes pruritus, typically characteristic of AD patients. The transient receptor potential vanilloid type 1 (TRPV1) is a cation channel activated by diverse noxious stimuli that has been studied in a variety of pruritic skin diseases. In this study, the AD animal model was generated by administering the hapten, trinitrochlorobenzene (TNCB), to Nc/Nga mice, and the degree of expression of the IL-31 receptor alpha (IL-31RA) and TRPV1 in the skin of these atopic models was evaluated. The Nc/Nga mice were divided into 3 groups: control, TNCB 2-weeks treated, and TNCB 8-weeks treated. After inducing AD, the skin lesions in each group were scored and compared, and the histology of the skin lesions and the IL-31RA and TRPV1 expression for each group were evaluated by analyzing immunohistochemistry. The results show a significant difference in the skin lesion scores between the groups. The immunohistochemistry evaluation highlighted the remarkable expression of IL-31RA and TRPV1 in the nerve fibers of the TNCB 8-weeks-treated group. We thus confirmed that the long-term application of TNCB induced chronic atopic-like dermatitis and that IL-31RA and TRPV1 were overexpressed in the peripheral nerve fibers in this AD model.

## 1. Introduction

Atopic dermatitis (AD) is a common chronic inflammatory skin disorder characterized by dry skin and intense pruritus. Even though the pathogenesis of AD has not been completely elucidated, it has been demonstrated that skin barrier disruption, dysregulation of the immune system, and chronic pruritus are critical components of the pathogenesis of AD [1,2].

The epidermis plays an important role as a physical and functional barrier, and defects in the skin barrier are significant pathologic findings in AD patients [1]. Numerous studies have suggested that a decrease in filaggrin (FLG), a key epidermal protein responsible for epidermal function, may lead to impaired skin barrier function, thereby increasing susceptibility to allergens and microorganisms [3]. Skin barrier dysfunction has been considered the first stage in the development of AD [4,5]. However, it is also now evident that allergic skin inflammatory reactions directly affect the skin barrier function by activating the type 2 immune response [6]. Damaged keratinocytes produce cytokines, one of which is thymic stromal lymphopoietin (TSLP), which proliferate naïve T cells to Th2 cells [7]. It has been observed that the expression of filaggrin decreases under the conditions of the Th2 immune response. Type 2 immune cytokines, such as interleukin (IL)-4 and IL-13 from Th2 cells, contribute to chemokine production, skin barrier dysfunction, and allergic inflammatory reactions [8,9]. Therefore, the relationship between skin barrier dysfunction and skin immune dysregulation is a vicious cycle amplified by feedback [10].

Pruritus is a predominant clinical symptom of AD and, unlike hives where itching occurs only because of histamine secreted from the mast cells, various mediators also cause itching in AD patients [11]. One of the mediators is IL-31, which is a major pruritogenic cytokine mainly produced by activated Th2 cells. IL-31 binds to a heterodimeric receptor complex comprising the IL-31 receptor alpha (IL-31RA) and oncostatin M receptor beta. Binding to this receptor complex subsequently activates some signaling pathways, resulting in severe pruritus in AD patients [12,13,14]. IL-31 signaling through Janus kinase (JAK)-1 is one of the major pathways that cause itching in AD [15]. Upadacitinib, which is recommended for the treatment of moderate to severe rheumatoid arthritis, is an oral selective JAK-1 inhibitor and has been recently recommended for the treatment of moderate to severe AD [16,17].

The transient receptor potential (TRP) channel protein, expressed mostly on the plasma membrane of various animal cell types, acts as a receptor that stimulates cells by causing an intracellular inflow of cations, mainly Ca^2+^. A wide variety of stimuli activate TRP, ranging from transmitter material to mechanical stimuli such as heat, chemicals, osmolality, and pain. Moreover, a single TRP channel can also respond to varying stimuli [18,19,20]. TRP channels in the skin are involved in epidermal homeostasis, inflammation, sensory function, and the biology of melanocytes and the hair by mediating calcium influx into the relevant cells [18]. Transient receptor potential vanilloid type 1 (TRPV1), belonging to the family of TRP channels, is a heat receptor channel responsive to high temperatures (>43 °C). It can also be activated by capsaicin, the major spicy ingredient in chili peppers. Because of this polymodal chemo-thermo sensitivity, the sensation of pain and heat can be experienced simultaneously when spicy food is consumed [18,21]. One study found that TRPV1 in neurons of the dorsal root ganglion plays an important role in acute and chronic inflammatory pain [22]. In the sensory nerves, TRPV1 activation releases neuropeptides, including substance P (SP) and calcitonin gene-related peptide (CGRP), which induce cutaneous neurogenic inflammation (CNI) [21].

There are few studies on IL-31 and TRPV1 expression in the skin of Nc/Nga mice, the mice model commonly used for AD. In this study, we examined the characteristics of AD in Nc/Nga mice with AD induced with trinitrochlorobenzene (TNCB) and evaluated the expression of IL-31 and TRPV1 in AD-induced skin lesions.

## 2. Results

### 2.1. Evaluation of Skin Lesions

Among the three groups, the TNCB 2-weeks-treated group had the highest skin lesion score, with acute lesions such as erythema/hemorrhage, ulceration, crusting, edema, excoriation/erosion, and scaling/dryness. The skin lesion scores were significantly different among the groups (*p* = 0.0015). The skin lesion scores of the TNCB 2-weeks-treated group (*p* = 0.0097) and the TNCB 8-weeks-treated group (*p* = 0.0097) were significantly different from those of the control group. The skin lesion scores of the TNCB 2-weeks-treated group were significantly different from those of the TNCB 8-weeks-treated group (*p* = 0.0112). The skin lesions of the TNCB 8-weeks-treated group showed mild to moderate erythema, edema, excoriation, scaling, and dryness (Figure 1).

### 2.2. Histological Analysis

Remarkable hyperkeratosis, epidermal hyperplasia, and inflammatory infiltration into the epidermis and dermis were observed in the skin lesions of the TNCB 2-weeks-treated group. Compared to the control group, mild epidermal hyperplasia and inflammatory infiltration of skin lesions were also observed in the TNCB 8-weeks-treated group (Figure 2a). Toluidine blue (TB) staining revealed an increased number of mast cells in the skin lesions of the TNCB 8-weeks-treated group. The number of mast cells observed in the TNCB 2-weeks-treated group was only slightly higher than that of the control group (Figure 2b).

### 2.3. Immunohistochemistry

Immunoreactivity to IL-31RA was not observed in the tissues of the control group. Compared to the control group, positive reactions to IL-31RA were detected in the nerve fiber of the dermis of the TNCB 2-weeks-treated group. The immunoreactivity to IL-31RA in the nerve fiber of the dermis was strongest in the TNCB 8-weeks-treated group among the three groups (Figure 3a).

In the control group, immunoreactivity to TRPV1 was observed mainly in the sebocytes surrounding the hair follicle. Interestingly, in the TNCB 8-weeks-treated group, in addition to the sebocytes, intense immunoreactivity to TRPV1 was also observed in the nerve fibers. Contrarily, immunoreactivity to TRPV1 was seen to be minimal in the TNCB 2-weeks-treated group (Figure 3b).

## 3. Discussion

To understand the pathology of AD, the Nc/Nga mouse model, which has genetic permeability barrier abnormalities [23] and develops AD in a conventional environment, was used as the experimental murine model [24]. However, the Nc/Nga mouse model has several disadvantages that include the requirement of special housing, the long time taken to induce AD, and the undependable elicitation of skin lesions. To overcome these disadvantages, a method of inducing AD by applying a hapten to Nc/Nga mice has previously been studied [25].

When a hapten is applied to the skin of the murine model, it initially shows a predominantly Th1-cell immune response similar to allergic contact dermatitis (ACD). Several studies have demonstrated that this model shifts from typical delayed-type hypersensitivity to more chronic dermatosis with complex features of AD [26]. Man et al. reported that inflammation induced using a single challenge with the hapten is sufficient to elicit ACD but not enough to induce abnormal epidermal function, a characteristic of AD. However, long-term hapten challenges distort ACD to become a chronic dermatosis, with features of AD, involving the development of skin barrier impairment and enabling more allergens to penetrate the skin [27]. Once the allergen breaches the barrier, the resulting allergen-induced inflammation can further damage the skin barrier. This amplifying procedure may play a key role in the exacerbation and perpetuation of AD [28]. One study also noted that the Th2 cytokine was more dominant when TNCB was applied long-term in Nc/Nga mice versus a short-term application [29]. Thus, repeated epicutaneous application of hapten for a long time drives Th1 with delayed-type hypersensitivity to become a Th2 immune response.

In this study, the TNCB 2-weeks-treated group had the most severe skin lesion scores, acanthosis, and infiltration of inflammatory cells. The TNCB 8-weeks-treated group also showed a stronger inflammatory response compared to the control group and showed remarkable infiltration of mast cells compared to the control group and the 2-weeks-treated group. IL-31 is predominantly produced by Th2 cells [12]. Additionally, in contrast to mild IL-31RA expression in the cutaneous nerve fiber of the TNCB 2-weeks-treated group, IL-31RA was significantly expressed in the cutaneous nerve fiber of the TNCB 8-weeks-treated group. Hence, the TNCB 8-weeks-treated group was considered to have atopic-like dermatitis where the Th2-type immune response was predominant.

IL-31RA, with the highest expression in the TNCB 8-weeks-treated group, is a receptor of the potent pruritogenic cytokine IL-31 [12]. The accurate mechanism of function of IL-31 in inducing pruritus is not yet completely understood, but it is thought to be due to an interaction between the immune system and the nervous system, as IL-31 and its receptor IL-31RA are highly expressed in the dorsal root ganglia of the cutaneous sensory nerve [13,30]. IL-31 acts as a pivotal link between the sensory nerve and the Th2-mediated immune system in the generation of T-cell-mediated itching. In addition, IL-31 may act directly on the peripheral nerve, causing the pruritus related to AD [31].

Several studies have suggested that IL-31 may become involved in a positive feedback loop in the development of skin inflammation because it can guide the secretion of proinflammatory mediators, which can consequentially stimulate inflammation by activating dendritic cells (DCs) [32]. Interestingly, a recent study demonstrated that IL-31 is only essential for the induction of pruritus, not for activating DCs or inducing inflammation in the contact dermatitis murine model induced with a hapten [33]. Furthermore, administration of anti-IL-31 or anti-IL-31RA results in a decline in scratching behavior but not in the severity of skin inflammation in the mouse model of AD [34]. The result of this study that IL-31RA was mildly expressed in the TNCB 2-weeks-treated group which had a more severe inflammatory response and was expressed remarkably in the TNCB 8-weeks-treated group, considering the model of atopic-like dermatitis, also supports the notion.

In a previous study, TRPV1 and phosphorylated TRPV1 were reported to be significantly increased in the AD-like lesions of Nc/Nga mice. The treatment of PAC-14028, a TRPV1 antagonist, alleviated scratching behavior and the degranulation of mast cells and IgG production [35]. Another study reported that the treatment of PAC-14028 administered to Nc/Nga mice improved skin barrier recovery by suppressing Th2 cytokines, indicating that TRPV1 contributes to the development of Th2-type dermatosis [35,36]. Although the precise mechanism of the development of the itching sensation has not been elucidated, recent studies have suggested that the effect of TRPV1 on the sensory nerve is significantly involved in the process. Histamine receptor 1 (H1R), which plays a major role in generating the histamine-induced itch, exerts its influence through TRPV1 activation. In TRPV1-deficient mice, histamine-induced intracellular calcium influx is attenuated, resulting in decreased scratching behavior [37,38]. IL-31 contributes to the chronic itch in AD [39], and TRPV1 activation is required for the IL-31-induced itching [13].

The TRPV1 in the sensory nerves is involved in conducting various sensations (including itching) as well as triggering CNI by stimulating the sensory nerves and releasing neuropeptides. The application of a TRPV1 agonist (capsaicin) to the sensory nerve releases SP, which induces local inflammation through mast cell degranulation, vasodilation, and extravasation of leukocytes [40]. Inflammatory mediators produced by local inflammation subsequently stimulate the sensory nerves to release neurotransmitters; this positive feedback of CNI contributes to chronic inflammation in human AD [41]. While in the present study, TRPV1 expression was observed only in the nerve fibers and sebocytes of the TNCB 8-weeks-treated group and in the sebocytes of the control group, TRPV1 has been found in earlier studies to be variously expressed in the keratinocytes, mast cells, and sebocytes of human skin [42]. Human keratinocytes are activated by capsaicin through TRPV1. The activated keratinocytes release proinflammatory mediators like prostaglandin E_2_ (PGE_2_) and IL-8 and induce cyclooxygenase-2 (COX-2) expression [43]. Thus, TRPV1 is involved in numerous inflammatory processes occurring in AD lesions.

Unfortunately, we could only confirm IL-31RA and TRPV1 expression in the nerve fiber of the dermis in this study. In further research, it would be necessary to also confirm IL-31RA and TRPV1 expression on the dorsal root ganglion.

## 4. Materials and Methods

### 4.1. Experimental Animals

Fifteen Nishiki-nezumi Cinnamon/Nagoya (Nc/Nga) mice were purchased (Central Laboratory Animal Inc., Seoul, Republic of Korea). All the mice were eight-week-old females and were housed under controlled environmental conditions (24 °C ± 2 °C and 55% ± 15% humidity, 12/12 h dark–light cycle). Additionally, water and food (LabDiet 5053, Orient Bio, Seongnam, Kyunggi-Do, Republic of Korea) were provided ad libitum. All protocols and guidelines were approved by the Kangwon National University Institutional Care and Animal Use Committee (KW-180705-4).

### 4.2. Inducing AD in NC/Nga Mice

On the first day of the experiment, the backs of all the mice were shaved with an electric clipper, and trinitrochlorobenzene (TNCB) (Sigma-Aldrich, St. Louis, MO, USA) was applied to the dorsal skin of the mice to induce AD. The control group was treated by applying 0.9% NaCl to the dorsal skin twice a week for 4 weeks. In the TNCB 2-weeks-treated group, 2% TNCB (100 μL) was applied to the dorsal skin of the mice 3 times a week for 2 weeks. In the TNCB 8-weeks-treated group, 2% TNCB (100 μL) was applied 3 times a week for 2 weeks, followed by the application of 0.2% TNCB (100 μL) 3 times a week for another 6 weeks (Figure 4).

### 4.3. Evaluation of Skin Lesions

Skin assessment was conducted once a week by two observers before the application of the reagent. The severity of the symptoms, such as erythema/hemorrhage, edema, excoriation/erosion, and scaling/dryness, was evaluated by scoring them as follows: 0 (none), 1 (mild), 2 (moderate), and 3 (severe). The total skin score was obtained as the sum of the scores for each item [44].

### 4.4. Histological Analysis

On the last day of the experiment, all the mice were over-anesthetized with Zoletil 50 (Virbac, Carros, France) and sacrificed by injecting 0.1 M phosphate-buffered saline (PBS) into the heart, followed by fixation with 4% paraformaldehyde in 0.1 M PBS. The dorsal skin tissue was subsequently resected and fixed in the same fixative, dehydrated with graded alcohol treatment, and cleared with xylene. The cleared tissues were embedded in paraffin and sectioned into 5 μm sections using a microtome (Leica Microsystems GmbH, Wetzlar, Germany). Each section was mounted onto silane-coated slides (Muto Pure Chemicals Co., Ltd., Tokyo, Japan). Hematoxylin and eosin (H&E) and toluidine blue (TB) staining was performed as a standard protocol. Pathologic changes, including inflammatory cell infiltration, skin cell hyperplasia, and keratinization in the skin tissues, were compared between each group. Mast cell density, expressed as the number of cells per 250 μm^2^, was evaluated for each section under a microscope.

### 4.5. Immunohistochemistry

The paraffin blocks were cut to 5 μm thickness, dewaxed in xylene, and rehydrated 7 times for 5 min each with descending graded alcohol. Endogenous peroxidase activity was blocked with 0.3% hydrogen peroxidase for 20 min. After washing in PBS, the sections were each incubated with polyclonal rabbit anti-TRPV1 (NB 100-1617; Novus Biologicals, Centennial, CO, USA; diluted 1:250 in PBS) and IL-31RA (bs-2631R, Bioss Antibodies, Boston, MA, USA; diluted 1:300 in PBS) at 4 °C for 2 days with the appropriate humidity. A biotinylated antibody (Goat Anti-Rabbit IgG Antibody Biotinylated, Vector Laboratories, Burlingame, CA, USA) was applied as the secondary antibody. The standard ABC staining method was carried out in the subsequent process. Between steps, sections were washed thrice with 0.01 M PBS for 10 min. Slides were analyzed under a microscope.

### 4.6. Statistical Analysis

All data were analyzed using the Mann–Whitney and Kruskal–Wallis tests performed using the GraphPad Prism (ver 7.04; GraphPad, San Diego, CA, USA) statistical analysis software, and *p* values less than 0.05 were considered to have statistical significance.

## 5. Conclusions

In conclusion, the current study confirmed that the long-term application of TNCB in Nc/Nga mice induced chronic atopic-like dermatitis, although the short-term application of TNCB induced severe inflammation rather than AD. Additionally, it was confirmed in the AD model that IL-31RA and TRPV1 were overexpressed in the peripheral nerve fiber. This result indicates that IL-31 and TRPV1 are important factors in the development of AD.

## Figures and Tables

**Figure 1 ijms-24-13521-f001:**
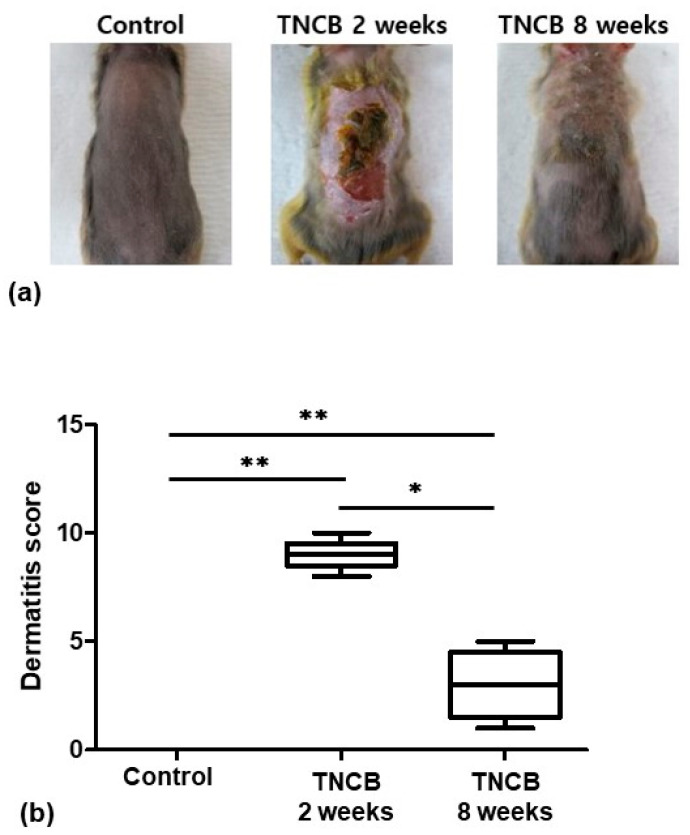
(**a**) Representative clinical features in the control, TNCB 2-weeks-treated, and TNCB 8-weeks-treated groups. (**b**) Dermatitis scores of the groups were significantly different. * *p* < 0.05; ** *p* < 0.01.

**Figure 2 ijms-24-13521-f002:**
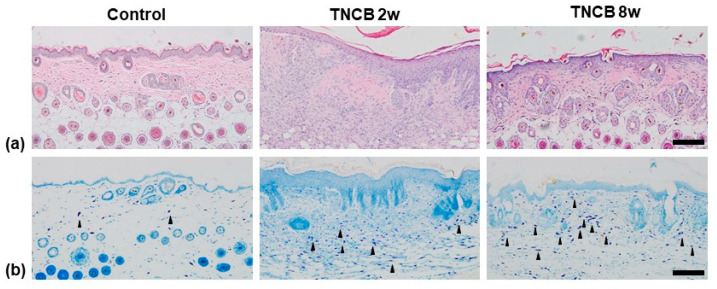
(**a**) Hematoxylin and eosin (H&E) staining. (**b**) Toluidine blue (TB) staining. Significant hyperkeratosis, epidermal hyperplasia, and inflammatory infiltration were observed in the TNCB 2-weeks-treated group. Increased number of mast cells (black arrowhead) was observed in the TNCB 8-weeks-treated group. Scale bar = 100 μm.

**Figure 3 ijms-24-13521-f003:**
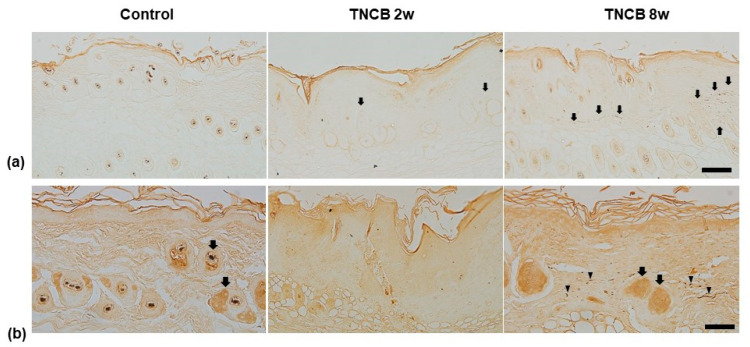
(**a**) Immunoreactivity to IL-31RA in skin lesions. In the TNCB 2-weeks-treated group, the nerve fiber (arrow) of the dermis was mildly immunoreactive to IL-31RA. In the TNCB 8-weeks-treated group, immunoreactivity to IL-31RA was also observed in the nerve fiber (arrow). (**b**) Immunoreactivity to TRPV1 in the skin lesions. In the TNCB 8-weeks-treated group, nerve fibers (arrowhead) and sebocytes (arrow) in the dermis were immunoreactive to TRPV1. In the control group, immunoreactivity to TRPV1 was only observed in the sebocytes. In the TNCB 2-weeks-treated group, there was little immunoreactivity to TRPV1. Scale bar = 100 μm.

**Figure 4 ijms-24-13521-f004:**
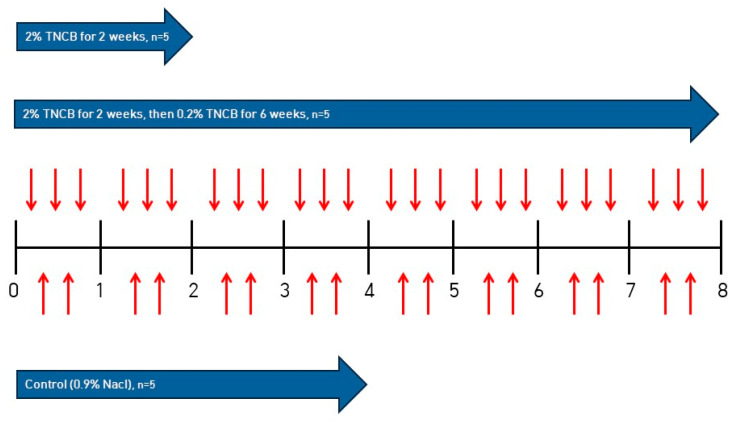
Scheme of AD induction in Nc/Nga mice.

## Data Availability

The data are contained within the article.

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
