# Peer review of "IL-31RA and TRPV1 Expression in Atopic Dermatitis Induced with Trinitrochlorobenzene in Nc/Nga Mice"

_ijms, 2023, doi:10.3390/ijms241713521_

Round 1

Reviewer 1 Report

Very interesting and well written article

I only have minor revisions

1) reverse the paragraphs, the material and methods paragraph should be inserted after the introduction

2) add paragraph limitations of the study

3) the introductory part should also refer to the treatment of atopic dermatitis in particare of jak inhibitors on the itching of the condition, I leave a very useful reference for the authors that they could use 

- DOI: 10.1111/jdv.18137

4) Minimal language review is required

5) There are some typos, proofread the whole text

Minimal language review is required

Author Response

Point 1) reverse the paragraphs, the material and methods paragraph should be inserted after the introduction.

Response: Thank you for your comment. However we arranged the sequence according to the instructions for authors.

Point 2) add paragraph limitations of the study

Response: We added the paragraph of limitation of the study as your comment (line 213-215).

Point 3) the introductory part should also refer to the treatment of atopic dermatitis in particare of jak inhibitors on the itching of the condition, I leave a very useful reference for the authors that they could use - DOI: 10.1111/jdv.18137.

Response: As your comment, we added the contents about the treatment of atopic dermatitis (line 59-63).

Point 4) Minimal language review is required.

Response: As your comment, the whole manuscript was proofread this time.

Point 5) There are some typos, proofread the whole text

Response: Sorry about the mistake. Now the whole manuscript was proofread.

Reviewer 2 Report

This is an interesting communication about IL-31RA and TRPV1 Expression in Atopic Dermatitis Induced by Trinitrochlorobenzene in NC/Nga mice. The article confirms the importance of IL31 and TRPV1 in the devolopment of AD. I think that the material and methods, the presented results, the discussion and conclusions are coeherent.

Author Response

This is an interesting communication about IL-31RA and TRPV1 Expression in Atopic Dermatitis Induced by Trinitrochlorobenzene in NC/Nga mice. The article confirms the importance of IL31 and TRPV1 in the development of AD. I think that the material and methods, the presented results, the discussion and conclusions are coherent.

Response: Thank you for your comment.